# Zuo1 supports G4 structure formation and directs repair toward nucleotide excision repair

Alessio De Magis [1,4], Silvia Götz[2,3,4], Mona Hajikazemi[1], Enikő Fekete-Szücs[3], Marco Caterino [1], Stefan Juranek [1] & Katrin Paeschke [1,2,3✉]

Nucleic acids can fold into G-quadruplex (G4) structures that can fine-tune biological processes. Proteins are required to recognize G4 structures and coordinate their function. Here we identify Zuo1 as a novel G4-binding protein in vitro and in vivo. In vivo in the absence of Zuo1 fewer G4 structures form, cell growth slows and cells become UV sensitive. Subsequent experiments reveal that these cellular changes are due to reduced levels of G4 structures. Zuo1 function at G4 structures results in the recruitment of nucleotide excision repair (NER) factors, which has a positive effect on genome stability. Cells lacking functional NER, as well as Zuo1, accumulate G4 structures, which become accessible to translesion synthesis. Our results suggest a model in which Zuo1 supports NER function and regulates the choice of the DNA repair pathway nearby G4 structures.

[1] Department of Oncology, Hematology and Rheumatology, University Hospital Bonn, Venusberg-Campus 1, 53127 Bonn, Germany. [2] Department of Biochemistry, Biocenter, University of Würzburg, Am Hubland, 97074 Würzburg, Germany. [3] European Research Institute for the Biology of Ageing, University of Groningen, A. Deusinglaan 1, 9713 AV Groningen, the Netherlands. [4] These authors contributed equally: Alessio De Magis, Silvia Götz. ✉email: katrin.paeschke@ukbonn.de

The demonstration that secondary DNA and RNA structures influence biological processes has revolutionized modern biology and brought attention particularly toward G-quadruplex (G4) structures. These are non-canonical secondary arrangements of (at least two) $\pi-\pi$ stacking guanine tetrads that form within guanine-rich DNA and RNA sequences[1,2]. While controversially discussed in the past, there is growing evidence of their formation and biological function in vivo, which is conserved from bacteria to human[3]. In yeast and human, G4 structure-forming sequences (G4 motifs) are significantly enriched at key functional units like promoters, mitotic and meiotic double-strand breaks (DSBs), and telomeres[4–6], pointing to a variety of critical cellular functions including transcription, cell-cycle regulation and telomere maintenance[7]. As G4 structures intervene in such a variety of biological processes they need to be properly regulated and unwound. A large number of proteins, mostly helicases, unfold G4 structures in vitro and in vivo[8]. Changes in G4 structure formation and unfolding can lead to replication fork stalling[9,10], accumulation of deletions/mutations[11–13], genomic copy number alterations and a high recombination frequency[6,14–19]. In model organisms (*Caenorhabditis elegans* and *Saccharomyces cerevisiae)* as well as in human tissue culture it has been shown that changes in G4 structure regulation lead to genome instability[10,20–23].

Although the underlying mechanisms have yet to be clarified, the formation of G4 structures is connected to DNA repair as indicated by the findings that many G4 structure-interacting proteins are linked to DNA repair processes[24–29]. BRCA1 and Rad51, as well as Ku80, have been shown to interact with G4 structures and function during either homologous recombination (HR) or non-homologous end-joining (NHEJ), respectively[25,26]. In addition to these canonical repair pathways, post-replicative repair proteins such as the translesion synthesis (TLS) protein Rev1[27,29,30] and the polymerase $\theta$[31] have also been linked to G4 structure formation. Furthermore, the helicases XPD and XPB, involved in transcription regulation and nucleotide excision repair (NER), have been shown to regulate G4 structures both in vitro and in vivo[32]. These studies underline the finding that G4 structures are prone to breakage and are a risk for genome stability. Contrarily, G4 structure-induced damage is also beneficial for the cell during class-switch recombination, antigenic variations or the repair of oxidized guanines[33–36]. These contrary findings demonstrate that there must be a subtle equilibrium between G4 structure-induced genome instability and G4 structure-promoted repair processes. Nevertheless, detailed knowledge on the impact of G4 structures on DNA repair is currently missing.

Based on the here presented data we speculate that G4 structures serve either as loading platforms for proteins involved in DNA repair or as bumps, which are slowing down the replication upstream of a lesion and thereby influencing the choice between different repair systems. We identify more than 100 candidate proteins that bind to G4 structures in *S. cerevisiae*; among these is Zuo1. By in vitro and in vitro experiments we reveal that Zuo1 supports G4 structure formation and contributes to genome stability by recruiting NER factors. Especially after UV damage, when more G4 structures form, Zuo1 function is essential to preserve genome stability. Zuo1 modulates G4 structure levels and acts as a molecular switch for the selection of the appropriate DNA repair pathway.

## Results

### Zuo1 binds to G-quadruplex structures in vitro.
We performed yeast one-hybrid (Y1H) screens with a G4 motif as a bait region to identify proteins that recognize G4 structures in vivo. In detail,

a G4 motif from chromosome IX (G4$_{IX}$; GGGTACGGTGGG TAATAAGGGGAAGGTATCGGG) was used as bait sequence (bait-G4) and was integrated upstream of a reporter gene (Aureobasidin A resistance gene) (Fig. 1a). The in vitro folding of G4$_{IX}$ into a parallel quadruplex was confirmed by circular dichroism (CD), with characteristic peaks at 243 and 264 nm, in 100 mM K$^+$ (Fig. 1b)[37]. We identified 157 potential G4 structure-interacting proteins using this approach (Supplementary Data 1). Among the identified proteins was Zuo1, a conserved eukaryote-specific, multifunctional J-protein present in the cytosol and nucleus[38,39]. Its published function in transcription and DNA repair[40,41] makes it a prime candidate to further address its biological function at G4 structures in the cell. To validate the specific interaction of Zuo1 with the bait-G4 structure we performed a

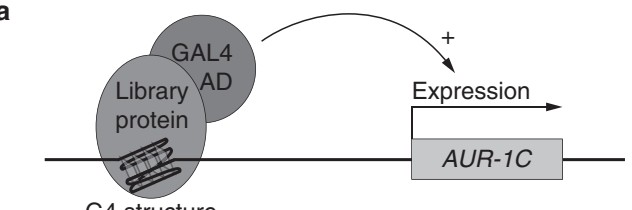

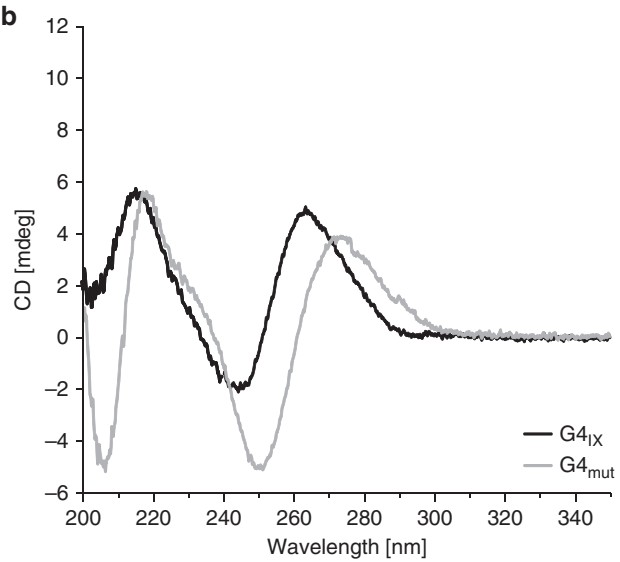

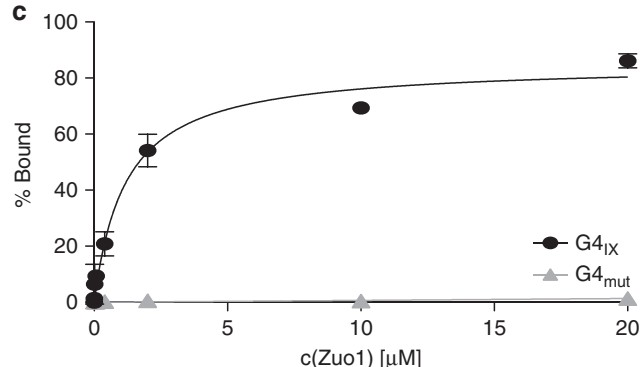

**Fig. 1 Zuo1 binds to G-quadruplex structures in vitro. a** Graphic illustration of the yeast one-hybrid screen (Y1H) system. GAL4 AD, Galactose Activation Domain. *AUR-1C*, Aureobasidin A **b** CD spectra of the folded oligonucleotides G4$_{IX}$ and G4$_{mut}$ in the presence of 100 mM K$^+$. **c** Quantification of Zuo1-binding to G4$_{IX}$ and G4$_{mut}$ by filter binding assay. Error bars correspond to one standard deviation of the mean of three independent experiments.

Y1H experiment using a mutated G4 motif ($G4_{mut}$) as a bait construct. CD analysis confirmed that no G4 structure forms within this mutated G4 sequence (Fig. 1b). The lack of growth on selective media when the mutated G4 motif was used as a bait indicated that Zuo1 binds specifically to the G4 sites in the Y1H assay.

There are two limitations in this approach: first, the interaction of Zuo1 with G4 structures can be direct or indirect; second, we cannot reveal whether Zuo1 binds to G4 structures or to unfolded G4 motifs. To overcome these restrictions, we purified Zuo1 from *Escherichia coli* (Supplementary Fig. S1a) and performed in vitro binding analyses (Fig. 1c). Zuo1-binding to G4 structures was determined by double-filter binding assays (Fig. 1c, Supplementary Fig. S1b–e) using four different G4 structures ($G4_{IX}$, $G4_{rDNA}$, $G4_{TP1}$, $G4_{TP2}$) and four non-G4 sequences as controls (dsDNA, $G4_{mut}$, forked and bubbled DNA). Double-filter binding analyses revealed that significant Zuo1 binding to all tested G4 structures (apparent $K_d$ range: 0.67–1.27 µM) and no binding to any control sequence (Fig. 1c, Supplementary Fig. S1b–e).

Furthermore, CD titration experiments under sub-optimal G4-stabilizing conditions (100 mM $Na^+$ in place of $K^+$) served to prove Zuo1 as able to influence the G4 conformational equilibrium. Under these conditions, $G4_{IX}$ indeed folds into a dominantly hybrid-1 quadruplex as seen by the CD spectrum with an additional distinct positive peak at 295 nm (Supplementary Fig. S1f). Increasing $Zuo1:G4_{IX}$ molar ratio prompted up to 14-fold ellipticity increase at 264 nm, along with the simultaneous decrease of the 295 nm band, proving the Zuo1-induced parallel $G4_{IX}$ stabilization.

**Zuo1 binds G4 motif sites genome-wide and supports G4 formation.** We performed chromatin immunoprecipitation (ChIP) followed by genome-wide sequencing analysis (ChIP-seq) in asynchronous yeast cultures expressing C-terminal Myc-tagged Zuo1 to test the binding of Zuo1 to G4 motifs in vivo. We obtained $6.1 \times 10^6$ reads of which 94% mapped to the *S. cerevisiae* genome (sacCer3). We identified 1594 chromosomal binding sites for Zuo1 using MACS 2.0 (Fig. 2a, Supplementary Data 2). Peaks were compared with genomic features (centromeres, ARS and promoters as annotated by SGD, https://www.yeastgenome.org), previously identified protein-binding regions (Pif1, γ-H2AX,

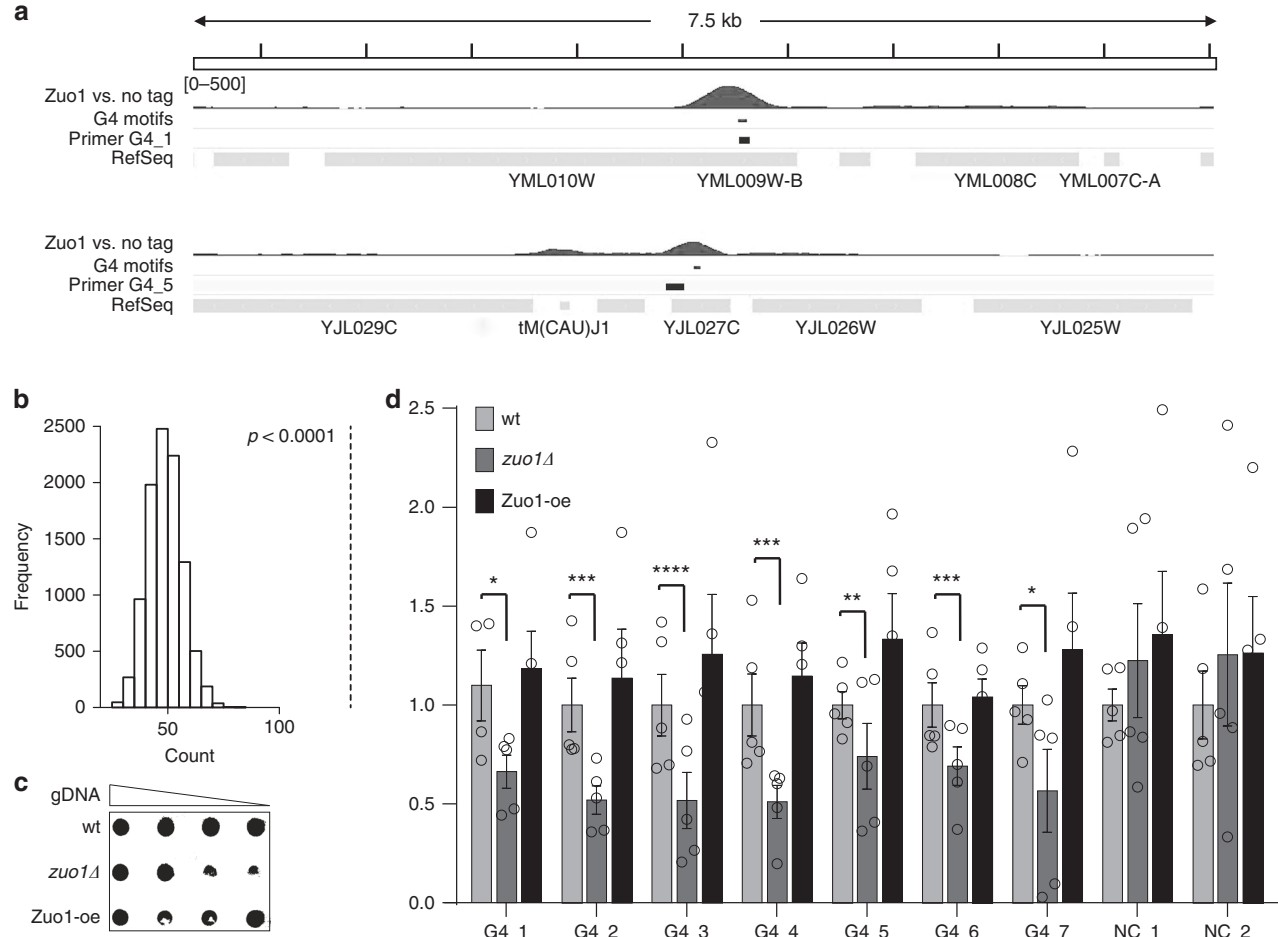

**Fig. 2 Zuo1 binds G4 motifs genome-wide and support G4 formation. a** IGV Genome Browser screenshot. Illustration of two 7.5 kb regions: top Chr XIII location 246,726 to 254,317 (G4_1) and bottom Chr X location 388,395 to 395,967 (G4_5). Zuo1-binding sites, location of G4 motifs and annotated genes are illustrated below. These regions were also among the regions tested in the qPCR, primers are indicated G4_1 and G4_5 (black track). **b** Genome-wide overlap of the Zuo1 peaks with G4 motifs as predicted by[4]. The black dotted line indicates a highly significant overlap $p = 0.0001$. **c** Southern–western combination to detect G4 structures in wildtype (wt), *zuo1Δ* and Zuo1-oe cells. Different amounts of genomic DNA were spotted on a membrane (2, 1, 0.5, and 0.25 µg), incubated with 2 µg/ml of BG4 antibody and detected by chemiluminescence. **d** BG4-ChIP analysis followed by qPCR of G4 levels in wt, *zuo1Δ*, and Zuo1-oe cells. Plotted are the means of $n = 5$ biologically independent experiments, which were normalized to wildtype. Error bars present ±SEM. Significance was calculated based on one-sided Student's *t*-test. Asterisks indicate statistical significance in comparison with wildtype. *$p < 0.05$, ** $p < 0.01$, ***$p < 0.001$, ****$p < 0.0001$.

DNA Pol2) and regions harboring putative G4 motifs[4,9]. Peaks significantly overlapped to G4 motifs (Fig. 2a, b), promoters ($p = 0.007$), replication pausing sites and R-loops ($p = 0.0001$)[42]. No correlation with DNA damage sites marked by phosphorylated H2Ax (γ-H2AX)[4] was observed (Supplementary Fig. S2a–d).

To test, whether Zuo1 changes the G4 structure level in the cell we analyzed the amount of folded G4 structures in Zuo1 deletion (*zuo1Δ*), Zuo1 overexpression (Zuo-oe) and wildtype cells. Genomic DNA was isolated, spotted at four concentrations on a nylon membrane and probed for G4 structures using the G4 structure-specific antibody BG4[43]. *zuo1Δ* showed ~50% less G4 structures than wildtype cells whereas no change could be determined in Zuo1-oe cells (Fig. 2c, Supplementary Fig. S2e).

Cellular G4 structure levels can also be measured by ChIP. We adapted the published protocol[44] to yeast and performed ChIP-qPCR. First, to validate the robustness of the method we monitored G4 structure levels in wildtype cells before and after the addition of PhenDC$_3$, an established G4-stabilizer[45]. We expected an increase of G4 structure levels after treatment with PhenDC$_3$. The ChIP-qPCR analyses confirmed that G4 structures form in vivo at selected sites (two- to three-fold enriched compared with the no antibody control) and more G4 structures were detectable after PhenDC$_3$ treatment (four- to eight-fold enriched) (Supplementary Fig. S2f). Here and in all subsequent ChIP and qPCR experiments we used seven Zuo1 target sites (G4_1 to G4_7), which overlap annotated G4 motifs[4], as well as two negative controls (NC_1, NC_2), which neither fold into G4 structures nor overlap with Zuo1-binding sites (see Supplementary Table S1 for qPCR primer).

We monitored G4 structures by ChIP in wildtype, *zuo1Δ* and Zuo1-oe cells. Similar to the previous experiment, a two-fold decrease in G4 signal was measured at all selected Zuo1 target sites in *zuo1Δ* cells (Fig. 2d). No significant changes in G4 structure levels were detected upon overexpression of Zuo1. We explain this by the finding that Zuo1 binds to a specific subset of G4 regions that do not increase upon Zuo1 overexpression. Meaning increasing amounts of Zuo1 do not increase the G4 targets that are bound by Zuo1. These data showed that Zuo1 binds to G4 structures and supports their formation.

**Zuo1 function at G4 has a positive effect on cellular fitness**. To understand the cellular role of Zuo1 and the underlying cellular processes, we monitored the cellular consequences of Zuo1 deletion. As the first sign of an unbalanced homeostasis cellular growth is impaired. Changes in cellular growth can be monitored in liquid or on plates. The doubling time of *zuo1Δ* cells increased to 144 min as compared with 90 min in wildtype cells (Fig. 3a, b, $p = 0.0003$). We induced G4 structure formation chemically by adding PhenDC$_3$ to wildtype and *zuo1Δ* cells (Supplementary Fig. S2f) to assess whether the observed growth defect (Fig. 3a) is due to reduced G4 structures in the cells (Fig. 2c, d). We monitored growth by spotting different concentrations of yeast cells on plates containing 10 μM PhenDC$_3$. Upon PhenDC$_3$ addition no changes in colony formation for wildtype cells was detected, but for *zuo1Δ* cells colony formation was increased indicating that G4 structure stabilization rescued the growth defect of *zuo1Δ* (Fig. 3b). In liquid media, we confirmed that PhenDC$_3$ treatment significantly rescues the growth defects of *zuo1Δ* (without PhenDC$_3$ 144 min, with PhenDC$_3$ 112 min doubling time, Supplementary Fig. S3a).

Pif1 and the RecQ helicase Sgs1 have been described to regulate G4 structures in yeast[5,9,13]. We therefore questioned whether Zuo1 interacts with known G4-unwinding helicases. To test whether Zuo1 functions in the same pathway as either Sgs1 or Pif1, we created *zuo1Δ sgs1Δ* and *zuo1Δ pif1-m2* yeast strains.

Cells with a specific point mutation in the *PIF1* gene (*pif1-m2*) lack the nuclear isoform of Pif1 but express the mitochondrial isoform[46]. Both *sgs1Δ* and *pif1-m2* do not have a growth defect[46,47]. The double mutants *zuo1Δ sgs1Δ* and *zuo1Δ pif1-m2* exhibited prolonged doubling time compared with *zuo1Δ* (Fig. 3c). Doubling times of 225.6 min for *zuo1Δ sgs1Δ* and 155.4 min for *zuo1Δ pif1-m2* were determined. This hints that Zuo1 does not act in the same pathway as Sgs1 and Pif1 because the double mutant would not increase the initial growth defect otherwise (Supplementary Fig. S3b,c shows the growth rates of single and double mutants). To test whether the growth defects in the double mutants are due reduced G4 structures, we stabilized G4 structures by adding PhenDC$_3$. The growth defect in *zuo1Δ sgs1Δ* was rescued after the re-stabilization of G4 structures, indicating that Sgs1 and Zuo1 functions are likely connected to G4 structures (Fig. 3d, Supplementary Fig. S3d). No growth changes were observed in *zuo1Δ pif1-m2* cells after PhenDC$_3$ addition. To test whether either Sgs1 or Pif1 binds to Zuo1 target regions and whether this binding depends on Zuo1, we monitored Pif1 and Sgs1 binding to seven Zuo1 targets and two control regions by ChIP-qPCR (see above). Pif1 did not bind significantly to Zuo1 targets and, consequently, its binding did not change in *zuo1Δ* (Fig. 3f). Sgs1 binding was four-fold reduced in the absence of Zuo1 (Fig. 3e). These results revealed that Zuo1 and Pif1 do not act in the same pathway and targets. However, these data demonstrated that Zuo1 is essential for Sgs1 binding to these G4 sites.

**Zuo1 mediates NER pathway recognition at G4 sites**. The published function of Zuo1 in transcriptional regulation[40,41] and the potential function of G4 structures at promoters, prompted us to investigate potential transcriptional changes between wildtype and *zuo1Δ* cells. In a microarray-based screen we identified 80 up- and 142 down-regulated genes in response to Zuo1 deletion. However, no direct correlation to Zuo1 targets could be determined (Supplementary Data 3, Supplementary Fig. S3e).

It has been shown that G4 structure formation can cause DNA damage and drive DNA damage response (DDR) activation[48,49] in the absence of helicases[13,50]. DSBs are life-threatening lesions in the genomic DNA repaired by HR or NHEJ[51]. To determine whether Zuo1 recruits either NHEJ or HR proteins to target G4 sites, we endogenously tagged yKu70 (NHEJ) and Rad50 (HR) with Myc13. ChIP-qPCR analysis in wildtype and *zuo1Δ* indicated that neither pathway is triggered at these sites nor is altered in a Zuo1-dependent fashion (Fig. 4a, b). Zuo1 binding was also not significantly altered in the absence of either Rad50 or yKuo70 (Supplementary Fig. S4a, b). These results agree with the lack of a significant overlap between Zuo1 targets and γH2ax loci (Supplementary Fig. S2b). Furthermore, rearrangement rates monitored in gross chromosomal rearrangement (GCR) assays at a G4-specific locus, did not show elevated rates in *zuo1Δ* cells either (Supplementary Fig. S4c). These results indicated that the Zuo1 function at G4 structures is not affecting DSB formation and canonical DNA repair.

*zuo1Δ* cells are sensitive to DNA damage agents such as UV, bleomycin, hydroxyurea (HU), or methyl methanesulfonate (MMS) (Supplementary Fig. S4d). This implies a post-replicative function of Zuo1. TLS, base excision repair (BER) and NER are prominent post-replicative repair pathways. We monitored whether the proteins of these pathways bind to Zuo1 targets and if such interactions depend on the presence of Zuo1. We endogenously tagged for each repair pathway one protein: Rev1 (TLS), Apn1 (BER) and Rad23 (NER). We analyzed the binding of the proteins in wildtype and *zuo1Δ* cells by ChIP-qPCR. ChIP-qPCR data were normalized to *zuo1Δ*/wildtype and fold decrease was plotted. Apn1 and Rev1 showed low levels of binding to Zuo1

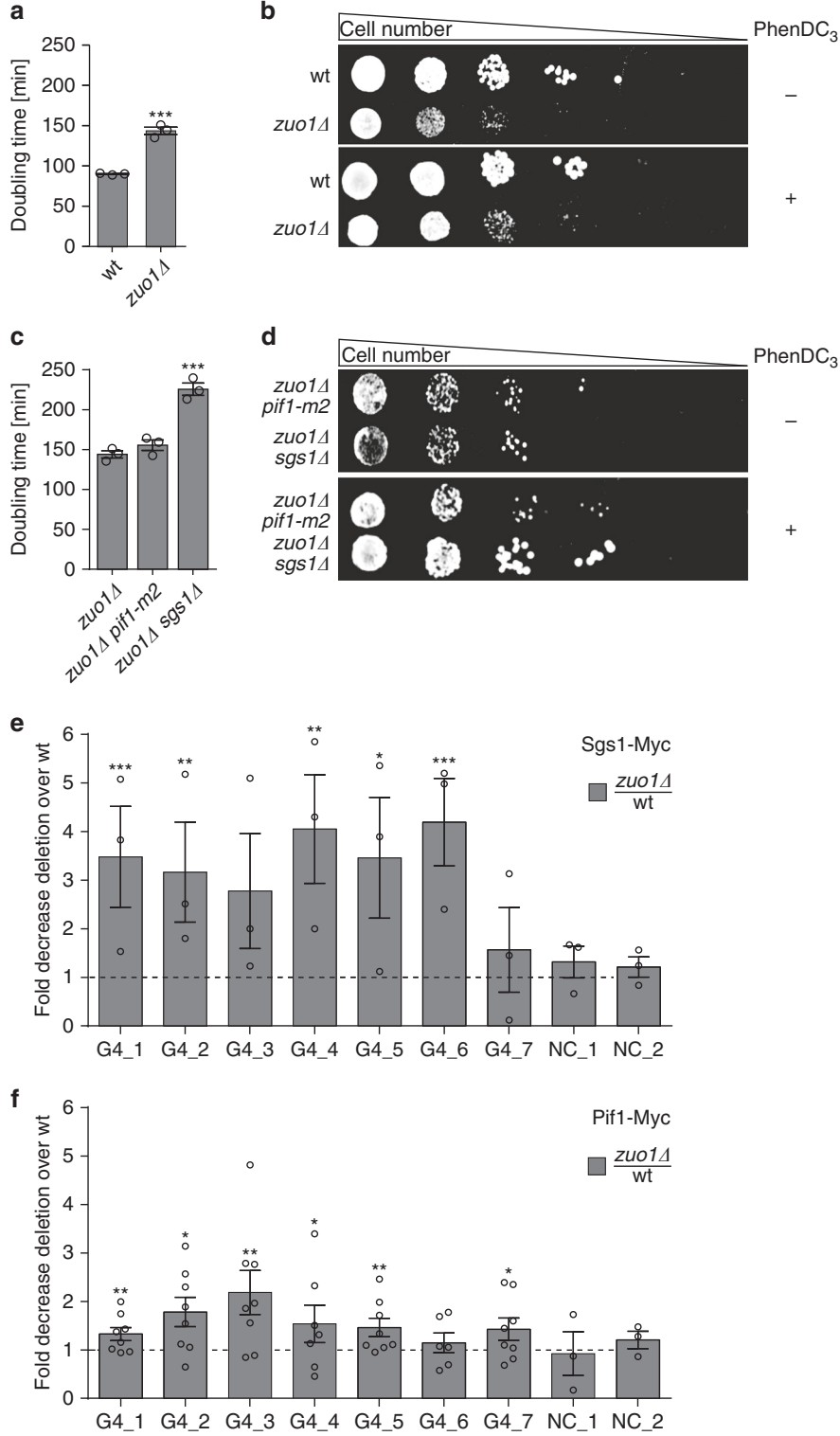

**Fig. 3 Zuo1 function at G4 has a positive effect on cellular fitness. a–d** Monitoring of growth changes in different yeast cells **a**, **c** Growth curves in liquid media were performed and doubling times (minutes) were calculated using the indicated yeast strains. Plotted are the means of n = 3 biologically independent experiments. Error bars present ±SEM. Significance was calculated based on one-sided Student's t-test. Asterisks indicate statistical significance: *p < 0.05, **p < 0.01, ***p < 0.001. **b**, **d** A serial dilution of yeast cells was spotted on rich media with and without 10 μM PhenDC₃. Growth changes and sensitivity of n = 3 biologically independent experiments were monitored by colonies formation. **e**, **f** ChIP analysis followed by qPCR that monitored the binding of either Sgs1 (**e**) or Pif1 (**f**) to seven Zuo1 targets (G4_1–7) and two controls (NC_1,2) was monitored after ChIP and qPCR. ChIP was performed in wildtype and zuo1Δ cells. Presented data show fold decrease zuo1Δ/wt ± SEM. For all experiments, the means of three biological replicates were plotted. Significance was calculated based on one-sided Student's t-test. Asterisks indicate statistical significance in comparison with wildtype cells. *p < 0.05, **p < 0.01, ***p < 0.001.

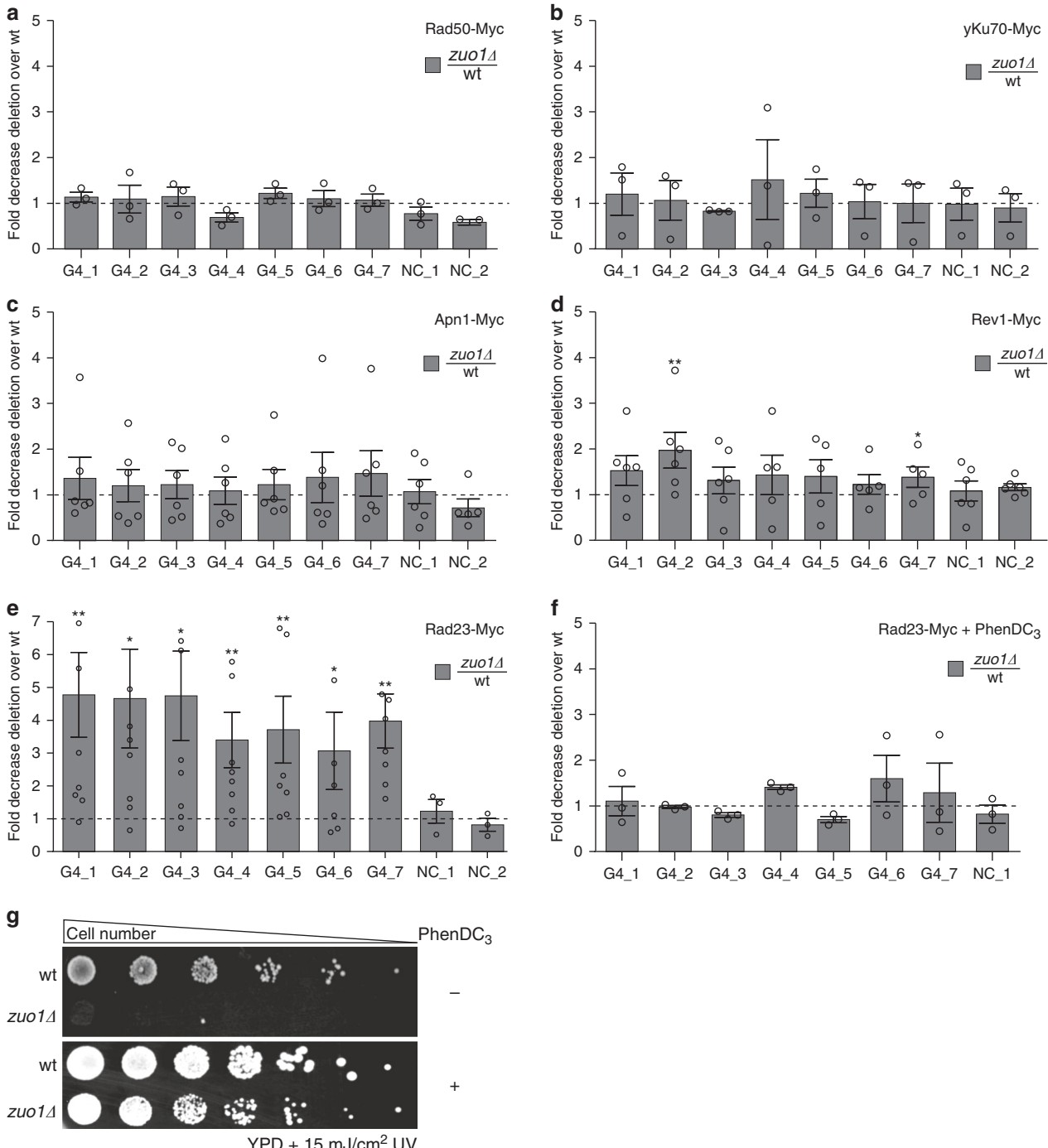

**Fig. 4 Zuo1 mediates NER pathway recognition at G4 sites.** ChIP and qPCR analysis of different repair proteins in wildtype and *zuo1Δ* cells. All qPCRs were performed at seven Zuo1 targets (G4_1–7) and two controls (NC_1, 2). Presented data show fold decrease zuo1Δ/wt ± SEM of $n = 3$ biologically independent experiments. **a** ChIP and qPCR of Rad50-Myc. **b** ChIP and qPCR of Ku70-Myc **c** ChIP and qPCR of Apn1-Myc **d** ChIP and qPCR of Rev1-Myc **e** ChIP and qPCR of Rad23-Myc **f** ChIP and qPCR of Rad23-Myc in the presence of 10 μM of PhenDC$_3$. All plotted results were based on the average of at least three independent experiments ± SEM. Significance was calculated based on one-sided Student's *t*-test. Asterisks indicate statistical significance in comparison with wildtype cells under the same experimental conditions. *$p < 0.05$, **$p < 0.01$. **g** Yeast cells were grown on liquid media in the presence or absence of PhenDC$_3$, irradiated with 15 J m$^{-2}$ UV light (254 nm) and spotted in different concentrations of rich media. Growth changes and sensitivity were monitored by colony formation.

targets and no changes in binding between wildtype and *zuo1Δ* (Fig. 4c, d) indicating that neither BER nor TLS acts at the G4 sites targeted by Zuo1. However, Rad23 (a subunit of the Rad4/Rad23 complex; XPC in human) changed its binding pattern to G4 sites in the absence of Zuo1. Zuo1 deletion resulted in at least a three-fold decrease in the binding of Rad23 to G4 sites (Fig. 4e). These results

indicated that Zuo1 supports the binding of Rad23 to G4 motifs. To exclude that this effect is specific to Rad23, we monitored the binding pattern of additional NER proteins: Rad4, Rad1 (XPF in human) and Rad2 (XPG in human) (reviewed in[52]). Similar to Rad23 also Rad4, Rad1 and Rad2 exhibit significantly reduced binding to Zuo1 target regions in *zuo1Δ* cells (Supplementary

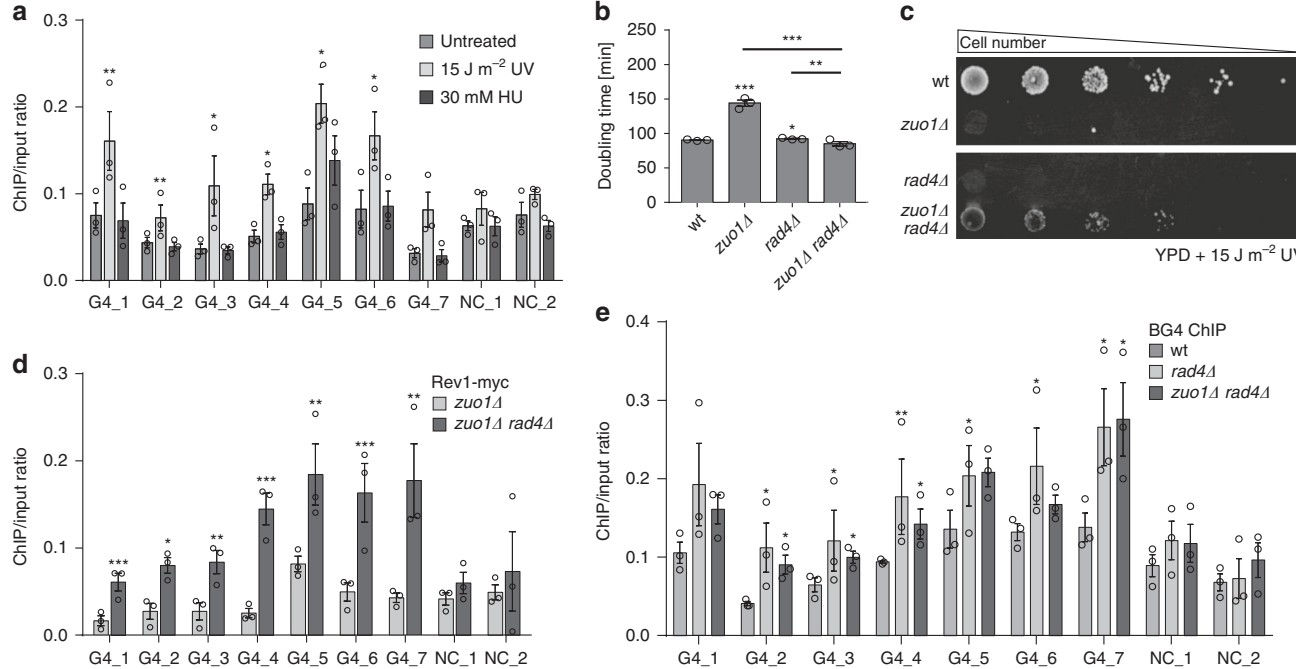

**Fig. 5 Zuo1 deficiency and NER impairment increase TLS activity at G4 sites. a** BG4-ChIP analysis followed by qPCR of G4 levels in untreated wildtype, treated with J m$^{-2}$ UV (254 nm) and 30 mM HU strains. Plotted are the means of $n = 3$ biologically independent experiments. Error bars present ±SEM. Significance was calculated based on one-sided Student's $t$-test. Asterisks indicate statistical significance: *$p < 0.05$, **$p < 0.01$, ***$p < 0.001$. **b** Growth curves of indicated yeast strains in liquid media. Doubling times (minu) were calculated. Plotted are the means of $n = 3$ biologically independent experiments. Error bars present ±SEM. Significance was calculated based on one-sided Student's $t$-test. Asterisks indicate statistical significance: *$p < 0.05$, **$p < 0.01$, ***$p < 0.001$. **c** Different concentrations of yeast cells were spotted on rich media, with or without irradiation with 15 J m$^{-2}$ UV light (254 nm). Growth changes and sensitivity were monitored by colony formation. **d** Rev1 Myc-ChIP analysis followed by qPCR to test Rev1 binding at nine different loci. The bars show the IP value over the input of the $zuo1\Delta$ strain as well as the double mutant $zuo1\Delta$ $rad4\Delta$. Plotted are the means of n = three biologically independent experiments. Error bars present ±SEM. Significance was calculated based on one-sided Student's $t$-test. Asterisks indicate statistical significance: *$p < 0.05$, **$p < 0.01$, ***$p < 0.001$. **e** BG4-ChIP analysis followed by qPCR of G4 levels in wildtype, $rad4\Delta$ and $zuo1\Delta$ $rad4\Delta$ strains. Plotted results are the means of $n = 3$ biologically independent experiments. Error bars present ±SEM. Significance was calculated based on one-sided Student's $t$-test. Asterisks indicate statistical significance in comparison with wildtype: *$p < 0.05$, **$p < 0.01$, ***$p < 0.001$.

Fig. S4e–g). To understand whether reduced G4 structure levels are causing the change in NER binding, we stabilized G4 structures by adding PhenDC$_3$ and measured the binding of Rad23 in wildtype and $zuo1\Delta$. ChIP-qPCR analyses showed that Rad23 binding in $zuo1\Delta$ cells is rescued after the addition of PhenDC$_3$ (Fig. 4f). This, as well as the finding that the UV sensitivity of $zuo1\Delta$ can also be rescued by PhenDC$_3$ addition (Fig. 4g), suggested that G4 structure stabilization itself recruits NER factors to bind and function at these sites.

**Zuo1 deficiency and NER impairment increase TLS activity at G4 sites.** Zuo1 binds and supports G4 structure formation leading to the recruitment of NER factors. Defects in NER[53–55], as well as the deletion of Zuo1, resulted in severe UV sensitivity (Fig. 5c, Supplementary Fig. S4d). In order to understand whether the UV sensitivity was due to a delay in the repair of UV lesions, we analyzed the levels of γH2AX wildtype and $zuo1\Delta$ cells after UV radiation over time by western blot. Thymine dimers and other photo adducts occurring upon UV irradiation lead to the recruitment of RPA, XPA, and XPC-TFIIH, hence to double-strand break processing[56]. Proteins of wildtype and $zuo1\Delta$ cells with UV were isolated at 0, 1, and 4 h after UV exposure. Western blot analysis demonstrated that, after 4 h, most DNA damages were cleared in wild type but not in $zuo1\Delta$. This data indicated a delay in eliminating UV lesions in $zuo1\Delta$ cells, which could explain the growth defects in $zuo1\Delta$ cells after UV treatment. Quantification of these western analyses revealed that without Zuo1 60% less DNA repair after UV damage occurs. We

speculated that G4 structures formed upon UV damage result in Zuo1-binding, which in turn facilitates NER recruitment. To test this hypothesis, we monitored G4 structure levels in wildtype cells upon UV treatment. In line with our assumption, at least two-fold more G4 structures are detectable by ChIP-qPCR upon UV damage compared with no treatment (Fig. 5a). This is specific to UV damage, because treatment with HU (replicative damage) did not increase G4 structure levels in the cells (Fig. 5a).

We monitored the growth in the double deletion $zuo1\Delta$ $rad4\Delta$ with the aim to characterize the relation between Zuo1 and the NER component Rad4. $zuo1\Delta$ exhibited a growth defect, whereas $rad4\Delta$ did not. Remarkably, $zuo1\Delta$ $rad4\Delta$ suppressed the growth defect of $zuo1\Delta$. The doubling time of $zuo1\Delta$ $rad4\Delta$ was 82 min, which is 57% faster than the single mutant $zuo1\Delta$ (Fig. 5b, Supplementary Fig. S5a). Both single mutants were UV sensitive (15 J m$^{-2}$), whereas the double mutant was not (Fig. 5c). These results led us to speculate that in the double mutant $zuo1\Delta$ $rad4\Delta$ an alternative repair pathway is recruited to compensate for the loss of the NER activity. To understand which DDR pathway is active in $zuo1\Delta$ $rad4\Delta$, we examined the binding of Rad50 (HR), Ku70 (NHEJ) and Rev1 (TLS) in $zuo1\Delta$ $rad4\Delta$ cells. ChIP-qPCR analyses were performed with these strains and confirmed that neither HR nor NHEJ compensates for the loss of Zuo1 and Rad4 at Zuo1 target regions (Supplementary Fig. S5b,c). However, Rev1 (TLS) showed at least a twofold increase in binding to Zuo1 target regions in $zuo1\Delta$ $rad4\Delta$ (Fig. 5d) compared with the single mutant $zuo1\Delta$. Defects in NER ($rad4\Delta$) alone were not sufficient to recruit Rev1 (Supplementary Fig. S5d).

To further examine this change in repair pathway and to connect this to G4 structure formation, we performed BG4 ChIP-qPCR analyses in wildtype, *zuo1Δ*, *rad4Δ*, and *zuo1Δ rad4Δ* cells. Again, less G4 structures were detectable in *zuo1Δ* compared with wildtype (Fig. 2c, d). In *rad4Δ* and *zuo1Δ rad4Δ*, significantly more G4 structures were detected compared with wildtype (Fig. 5e). These results confirmed that Zuo1 supports G4 structure formation, which stimulates the recruitment of NER components. In addition, a functional NER pathway is required for G4 unfolding. G4 structures accumulated and were accessible to TLS in cells lacking both functional Zuo1 and the NER machinery, as indicated by Rev1-binding. The activation of TLS in *zuo1Δ rad4Δ* cells did not make the cells sensitive to UV radiation, unlike the single mutants (Fig. 5c).

## Discussion

A number of studies link G4 structure formation to genome instability[9,12,14,20,22,28–30,50,57–62]. G4 structure formation has also been shown to positively influence biological processes such as telomere maintenance and transcription regulation[33,63–65]. Proteins that recognize and/or induce the formation of G4 structures are therefore required. Here, we identified 157 G4 structure-binding proteins by a Y1H screen. Among these is Zuo1 and we could show that it supports genome stability by assisting the recruitment of the NER machinery through binding and promoting the formation of G4 structures in *S. cerevisiae*.

*zuo1Δ* cells are sensitive to all tested DNA damaging agents (Supplementary Fig. S4d), which indicates that Zuo1 functions in post-replicative DNA repair. All post-replicative DNA repair processes (BER, TLS, and NER) are connected to G4 structure formation. During BER, G4 structure formation has been suggested to be stimulated by ROS-mediated oxidation of DNA and APE1 binding, which results in changes in transcription[33,34]. In eukaryotes, the polymerases Rev1, η, κ, and θ are involved in the replication of G4 motifs during TLS (reviewed in[66]). The helicases XPB and XPD of the NER pathway have been shown to act at G4 sites by ChIP-seq[32]. However, during post-replicative DNA repair, as well as during canonical DNA repair mechanism, G4 structure formation has been treated as the cause of the activation of the repair machinery[9,10]. Contrary, our data demonstrated that G4 structures targeted by Zuo1 do not lead to genome instability but rather support genome stability by recruiting repair factors to nearby lesions after UV damage (Figs. 2–5 and Supplementary Figs. S3, S4).

In detail, we showed that Zuo1 binding stimulates G4 structure formation (Fig. 2). However, these Zuo1-bound G4 structures did not lead to the recruitment of proteins of the HR or NHEJ machinery (Fig. 4), caused increased GCR rates (Supplementary Fig. S4c) or changed DNA replication fork progression (data not shown). Contrary, our data indicated that G4 structures formed and bound by Zuo1 positively supported the binding of the proteins of the NER machinery and contributed to NER function (Figs. 4, 5). The binding of Zuo1 to G4 structures was essential for NER function given the severe growth defect and UV sensitiveness of *zuo1Δ* cells (Figs. 2, 3 and Supplementary Fig. S4d).

The *zuo1Δ* phenotype could be unambiguously linked to the reduced cellular G4 structure levels because both the cellular doubling time and UV sensitivity could be rescued by treating *zuo1Δ* cells with the G4-stabilizer PhenDC₃ (Figs. 3b, 4g). In addition, PhenDC₃ also rescued the recruitment of NER machinery in *zuo1Δ*, as indicated by Rad23 binding (Fig. 4f). These data demonstrated that Zuo1 and G4 structure formation and function are mechanistically related and positively influence NER.

After UV irradiation, we observed an enrichment in G4 structure formation compared with wildtype (Fig. 5a). We argue that UV-induced G4 structures are recognized by Zuo1, which stabilizes these structures and facilitates the recruitment of NER proteins. The here presented data indicate that the function of Zuo1 at G4s is direct and not due to Zuo1 blocking the G4 regions against helicase function. Because neither the binding of Pif1 nor Sgs1 helicases are increased in Zuo1 deficient cells (Fig. 3).

These findings are in agreement with recent data showing ZRF1, the human orthologue of Zuo1, directly interacting with the NER machinery[67,68]. Although the function of ZRF1 is not clear, yet, it is conceivable to expect similarities with Zuo1 in supporting G4 structures and NER recruitment. Indeed, it has been shown that *zuo1Δ* growth defect can be rescued by expressing the human orthologue ZRF1[69]. Interestingly, the NER complex component Mms1 (DDB1 in human) can bind to G4 structures[58], further underlining the importance of G4 formation for NER function.

In Fig. 3 we showed that not only the *zuo1Δ* growth defect was rescued by the re-stabilization of G4 structures by PhenDC₃, but also that the binding of Sgs1 was again detectable upon G4 structure stabilization. This indicated that also Sgs1 binding to G4 structures is dependent on Zuo1 function at G4 structures. Sgs1 is a multifunctional helicase that belongs to the RecQ helicase family, which function is tightly connected to genome stability (reviewed in[70]). Defects in Sgs1 have been shown to be linked to defects in HR. Recently, it was shown that RecQ helicases also support NER in a so far unknown manner[71–75]. Sgs1 also interacts with the NER protein Rad16[76]. Combining these findings with our data (Fig. 3) we conclude that Sgs1 is recruited to Zuo1 target regions because of the presence of G4 structures. Without Zuo1, fewer G4 structures form (Fig. 2) and consequently the need for Sgs1-binding and function is reduced. Further analyzes are required to address the question of which function Sgs1 has at G4 sites during NER. A likely scenario is that Sgs1 unfolds G4 structures after NER has repaired the lesion.

In summary, our data lead to a model in which G4 structures have a positive effect on DNA repair (Fig. 6). We propose that, upon UV damage, Zuo1 is recruited at lesion sites by the formation of G4 structures. This results in the stabilization of the G4 structures in the vicinity of this lesion. This G4 stabilization stimulates the binding of the NER machinery, which results in efficient repair. Without Zuo1, less G4 structures form and the binding of NER proteins is reduced (Fig. 4). We draw the conclusion that, in the absence of Zuo1, NER is still acting at such sites but less efficiently, because *zuo1Δ* cells were UV sensitive and no other repair pathway was upregulated in *zuo1Δ*. The binding and function of NER components at G4 sites in *zuo1Δ* was underlined by the finding that without NER more G4 structures formed (Fig. 5), which in turn suggested that the NER machinery itself was involved in G4 structure unwinding. A potential candidate for this unwinding could be Sgs1 (Fig. 3). In the absence of Zuo1 and without an intact NER machinery (double deletion of *zuo1Δ* and *rad4Δ*) cells grew similar to wildtype and were no longer as UV sensitive (Fig. 5). This rescue of UV sensitivity can be explained by our finding that Rev1, the major protein involved in TLS, bound to Zuo1 target regions and compensated for the loss of Zuo1 and Rad4, most likely by repairing the lesion by TLS (Fig. 5). Rev1 bound also to G4 structures because in *zuo1Δ rad4Δ* cells more G4 structures formed in comparison to wildtype and both single mutants (Fig. 5e). This indicated that Zuo1 is not only a signal for NER but also prevents TLS at these sites. Furthermore, our findings demonstrate that G4 structures in the cell are important to assist the choice of DNA repair pathway in the vicinity of G4 structures.

## Methods

**Strains, constructs, and media**. All yeast strains are listed in Supplementary Table S2. All the strains used in this work are derivatives of the *RAD5* + version of

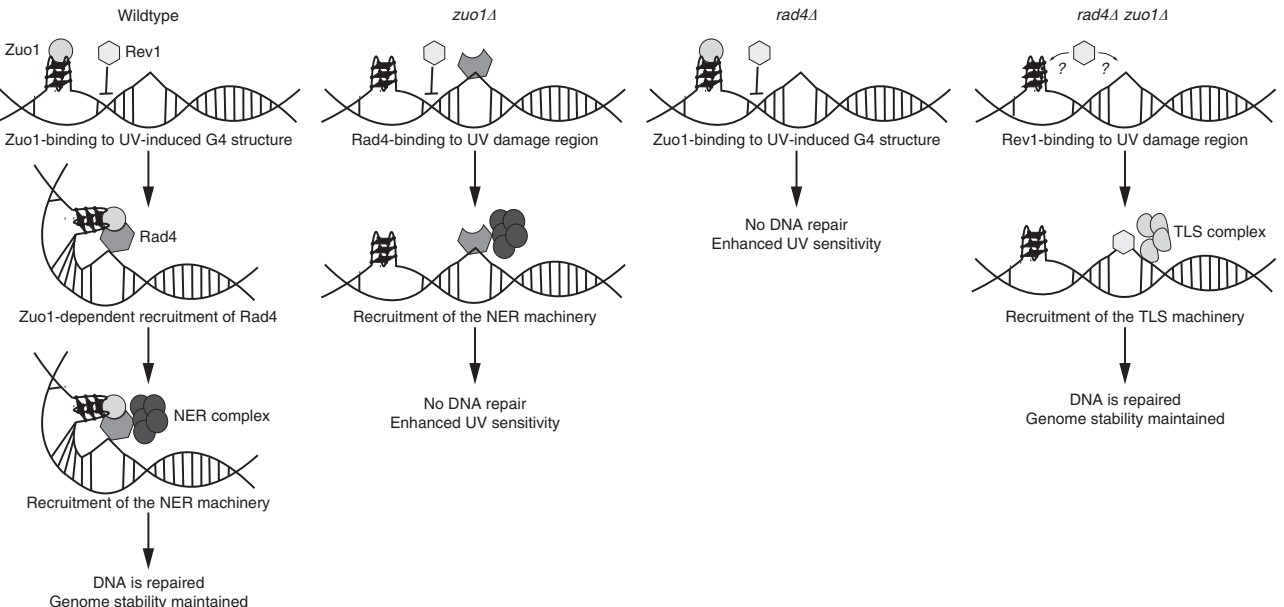

**Fig. 6 Illustration of Zuo1 function in G4 structure recognition, stabilization of DNA damage sites and the recruitment of the NER machinery.** In detail, UV exposure induces thymine dimer formation in the double helix. As a consequence, more G4 structures are formed that are supported by Zuo1 in the vicinity of the lesion. This stabilization stimulates the binding of the Rad4/23 complex. In the meantime, Zuo1 blocks the binding of Rev1 to those sites. The Rad4/23 complex recruits the NER machinery to repair the lesion and restore the double helix. Without Zuo1, G4 structures are not protected and cells are UV sensitive and the NER machinery is not recruited to the UV lesion. Only in the absence of both Zuo1 and a functional NER machinery an alternative pathway (TLS) is free to bind to these sites and repair the lesion.

W303 (R. Rothstein) or YPH background[77]. Deletions eliminated entire ORFs and were created using the pRS vector system[77]. Tagging at the endogenous locus with 13 Myc epitopes was performed by PCR using the pFA6A vector system[78]. Tagged proteins were expressed from endogenous loci and promoters. The *pif1-m2* point mutation was created by the pop-in/pop-out method using the pRS vector system[46].

**Yeast one-hybrid screen**. The yeast one-hybrid screens were performed using the Matchmaker™ Gold Yeast One-Hybrid Library Screening System (Clontech). A G4 motif from chromosome IX ($G4_{IX}$) with short flanking regions was cloned into the *S. cerevisiae* Y1HGold genome as described in the manual to construct the screening bait G4 strain. The control bait $G4_{mut}$ was cloned using the same strategy. After determination of the minimal inhibitory concentration of Aureobasidin A (AbA), screens were performed using the *S. cerevisiae* DUALhybrid cDNA library (Dualsystems Biotech). 7 µg cDNA library plasmid were transformed into the screening strain bait-G4 according to the manufacturer's protocol. After streaking out each yeast colony twice on selective plates the library plasmids were isolated from overnight cultures. Lysis was performed using DNA lysis buffer (2% (v/v) Triton X-100, 1% (w/v) SDS, 100 mM NaCl, 10 mM Tris HCl pH 8.0, 1 mM EDTA) and glass beads in a FastPrep instrument (MP Biomedicals™ FastPrep-24™) for 1 min at 4 °C, followed by phenol-chloroform extraction and ethanol precipitation. Plasmids were transformed in *E. coli* (XL-1 Blue) and overnight cultures were used to isolate plasmids by alkaline lysis. The obtained library plasmid was sent for sequencing using the primer GAL4ADseq (sequence from Dualsystems Biotech): 5′-ACCACTACAATGGATGATG-3′.

**Cloning, expression, and purification of Zuo1**. *Zuo1* was amplified by PCR from *S. cerevisiae* genomic DNA using these primers:
  SG117 (5′-AAAAAA*gaattc*ATGTTTTCTTTACCTACCCTAAC-3′),
  SG118 (5′-AAAAAA*gcggccgc*TCACACGAAGTAGGACAACAAG-3′).
  *Zuo1* was cloned into the *EcoR*I and *Not*I sites of a pET28a vector (Novagen). The resulting construct was confirmed by sequencing. 6 x His-tagged Zuo1 was expressed in Rosetta pLysS cells grown in LB medium supplemented with 25 µg/ml kanamycin (Applichem) and 30 µg ml⁻¹ chloramphenicol (Applichem), using 1 mM isopropyl β-D-thiogalactoside (IPTG, Applichem) for induction at 18 °C overnight, following the manufacturer's protocol and established protocols[79].
  All purification steps were carried out at 4 °C. Cell lysis was performed in lysis buffer (300 mM NaCl, 20 mM HEPES pH 7.5, 10% (v/v) glycerol, 1 mM DTT, 5 mM imidazole) using an EmulsiFlex-C5 homogenizer (Avestin). The supernatant of centrifuged cell lysate was applied onto a Ni-NTA agarose column (Thermo Scientific) pre-equilibrated with lysis buffer by gravity flow. After three washing steps with 1 column volume wash buffer (300 mM NaCl, 20 mM HEPES pH 7.5, 10% (v/v) glycerol, 1 mM DTT, 15 mM imidazole) bound protein was eluted with elution buffer (300 mM NaCl, 20 mM HEPES pH 7.5, 10% (v/v) glycerol, 1 mM

DTT, 250 mM imidazole). Zuo1-containing fractions were identified by 15% SDS-PAGE and western blotting with an anti-His antibody. Buffer of combined fractions was exchanged to lysis buffer without imidazole and the protein was concentrated using an Amicon Ultra-15 Centrifugal Filter Unit (MWCO 30 kDa). The protein concentration was measured by a Bradford assay and also determined by SDS-PAGE in comparison to known amounts of bovine serum albumin (BSA, Applichem) as a standard protein.
  The concentrated Zuo1-containing sample was subjected to a Superdex 200 (GE Healthcare) column and eluted with buffer (300 mM NaCl, 20 mM HEPES pH 7.5, 10% (v/v) glycerol, 1 mM DTT). BSA and aldolase were used as standard proteins for gel filtration.

**In vitro folding and analysis of G4 structures**. Oligodeoxynucleotides with a G4 motif were dissolved in buffer containing 100 mM KCl. After boiling G4 formation was induced by slowly reducing the temperature to room temperature[80]. G4 structure formation was confirmed by 7% SDS-PAGE and CD measurements. Oligodeoxynucleotides for control DNA structures[81] were treated likewise (Supplementary Table S3). Annealing was performed in annealing buffer (50 mM HEPES, 2 mM magnesium acetate, 100 mM potassium acetate) for 1 min at 98 °C, 60 min at 37 °C and 30 min at 22 °C. G4 structures and annealed control DNA structures for binding studies were desalted using illustra MicroSpin G-25 columns (GE Healthcare).

**Binding studies**. 20 pmol DNA was 5′-labeled with 25 µCi [γ-³²P] ATP using T4 polynucleotide kinase (NEB). G4 and G4mut structures were purified by 7% SDS-PAGE. Control DNA (ds, bubble, fork, 4 fork) was purified using illustra MicroSpin G-25 columns. DNA-protein-binding was analyzed by double-filter binding assays[82] using a 96-well Bio-Dot SF apparatus (Bio-Rad) and 10 nM DNA in binding buffer (50 mM Tris HCl pH 8.0, 125 mM KCl, 5 mM DTT, 10% (v/v) glycerol)[81]. Protein concentrations increased from 0 to 20 µM Zuo1. After incubation on ice for 30 min the reactions were filtered through a nitrocellulose and a positively charged nylon membrane, followed by three washing steps with binding buffer with no glycerol. The membranes were dried and analyzed by phosphoimaging on a Typhoon FLA 7000 (GE Healthcare). Percentage values of bound Zuo1 were determined using ImageQuant and were used to obtain dissociation equilibrium constants (apparent $K_d$) by curve fitting using nonlinear regression (Prism, Graphpad). The sequences of oligonucleotides used in these studies are listed in Supplementary Table S3.

**Circular dichroism (CD) spectroscopy**. CD spectra were recorded on a Jasco J-810 spectropolarimeter at 20 °C and data averaged over three scansions[58]. Oligos were dissolved in 100 mM KCl, 10 mM Tris HCl pH 7.0 buffer and annealed overnight after denaturation at 95 °C, 5 min. $G4_{IX}$ for CD titration was dissolved in 100 mM NaCl and 10 mM Tris HCl pH 7.0 buffer, heated at 95 °C for 5 min and quickly annealed on ice. Zuo1 was titrated against 2 µM DNA at 1, 2.5, and 5 Zuo1:

G4 molar ratios and spectra were recorded after 30 min incubation on ice. Oligonucleotide extinction coefficients were obtained by the nearest-neighbor method and concentration determined at 95 °C. Zuo1 concentration was determined using $\varepsilon_{280} = 4.641 \times 10^4\,M^{-1}\,cm^{-1}$.

**Myc-ChIP**. Myc-ChIP experiments were performed similar to previous published protocols[9]. Briefly, cells were lysed using glass beads in a Fastprep-24 and the chromatin was sheared to 200–1000 bp using a Bioruptor® Pico (Diagenode) with these settings: high intensity, 30 s ON, 30 s OFF, 7 cycles. Shearing quality was assessed on an 1% agarose gel. 8 µg anti-Myc antibody (Takara) was added to the sheared chromatin and incubated for 2 h at 4 °C followed by an incubation with 80 µl Dynabeads-Protein G (Thermo Scientific) for 2 h at 4 °C. After washing three times with washing buffer (100 mM KCl, 0.1% (w/v) Tween-20, 1 mM Tris HCl pH 7.5) the bound DNA was immunoprecipitated and analyzed by quantitative PCR (qPCR) using primers indicated in Supplementary Table S1.

**BG4-ChIP**. Cells were crosslinked and lysed and DNA was sheared similar to the Myc-ChIP protocol. 0.5 µg of BG4 antibody was added to 1 µg of sheared chromatin (resuspended in ChIP lysis buffer containing 1% (w/v) BSA) and incubated for 2 h at 16 °C followed by incubation with 40 µl FLAG M2 Magnetic Beads (Sigma) for 2 h at 16 °C. Beads were washed three times with washing buffer (100 mM KCl, 0.1% (w/v) Tween, 1 mM Tris HCl pH 7.5). Immunoprecipitated DNA was treated with Proteinase K at 37 °C and the crosslink was reversed at 65 °C for 2 min followed by overnight incubation at 16 °C.

Immunoprecipitated DNA was purified (PCR purification kit, Qiagen) and used for subsequent qPCR analyses. qPCR was performed using the iTaq Universal SYBR Green Supermix (BioRad). Fold enrichment of binding regions was quantified using the IP/Input method normalized to non-specific binding values. Microsoft Excel was used to plot the graphs and $p$ values were calculated using Student's $t$-test.

**ChIP-seq analysis**. Myc-ChIP experiments were performed as described above. For genome-wide sequencing DNA was treated according to manufacturer's instructions (Next ChIP-seq Library Prep Master Mix Set for Illumina, NEB) and submitted to deep sequencing (HiSeq 2500 sequencer). Obtained sequence reads were aligned to the yeast reference genome (sacCer3) with bowtie[83]. Binding regions were identified by using MACS 2.0 with default settings for narrow peaks[84]. Supplementary Data 1 contains all obtained Zuo1 peaks. The ChIP input was used as a control data set. Overlap of binding sites with other genomic features and binding regions were determined using a PERL script based on a permutation analysis between the query and subject features.

**Growth assay**. The strains used for growth assays are listed in Supplementary Table S2. Overnight cultures of *S. cerevisiae* strains were inoculated in YPD media to a starting OD (660 nm) of 0.1. Cultures were grown at 30 °C until an OD (660 nm) ≥ 1 was reached. Measurements were taken at 60 min intervals and doubling times were calculated from log phase OD (660 nm) values. Growth curves were performed in triplicates.

**Spot assay**. Yeast cultures were inoculated at OD (660 nm) of 0.15 using stationary *S. cerevisiae* culture and grown at 30 °C until OD (660 nm) 0.8 was reached. All yeast cultures were diluted to OD (660 nm) 0.8 and dilution series with six 1:5 dilutions were prepared in a 96-well plate. From each dilution, 3 µl were spotted on a plate and, after drying, incubated at 30 °C. After 2 days the plates were scanned and the growth of strains on different media was compared with estimate the growth defects. 10 µM PhenDC₃, 20 ng/ml Bleomycin (Calbiochem), or 100 mM HU (Sigma) was added to the medium to perform growth assays under G4-stabilizing and DNA damage conditions.

**BG4 purification**. The plasmid expressing an engineered antibody specific to G4 structures (BG4)[43] was kindly provided by S. Balasubramanian (University of Cambridge, UK). The plasmid was transformed into BL21(DE3) competent cells. Competent cells containing the plasmid were grown in 2XTY media (1.6% (w/v) bacto tryptone, 1% (w/v) bacto yeast extract and 0.5% (w/v) NaCl) and 50 µg ml⁻¹ kanamycin. Pre-culture was expanded in eight ×250 ml at OD (600 nm) of 0.1. At OD (600 nm) of 0.5 BG4 antibody expression was induced with 0.5 mM IPTG (isopropyl β-D-1-thiogalactopyranoside) at 25 °C for 16 h. The cells were lysed in TES buffer (50 mM Tris-Cl pH 8.0, 1 mM EDTA and 20% sucrose) on ice for 10 min. The lysate was diluted fivefold in water and left on ice for further 10 min prior to centrifugation at 10,000 $g$ at 4 °C for 30 min. The supernatant was filtered (0.2 µm) and purified on a Ni-NTA agarose (Thermo Scientific) column pre-equilibrated with TES buffer by gravity flow. The column was washed with PBS pH 8.0 containing 10 mM imidazole and BG4 antibody was eluted in PBS pH 8.0 containing 250 mM imidazole (pH was adjusted after imidazole addition). Imidazole-containing PBS was exchanged with inner cell salt buffer (25 mM Hepes (pH 7.6), 110 mM KCl, 10.5 mM NaCl, 1 mM MgCl₂). BG4 antibody was concentrated using an Amicon Ultra-15 Centrifugal Filter Unit (Millipore). BG4 antibody was quantified on a NanoDrop spectrophotometer

(Thermo Scientific) and stored at −80 °C. Purity of the BG4 preparation was monitored by SDS-PAGE.

**BG4 filter binding assay**. Asynchronous cultures were grown to OD (660 nm) of 0.6 and crosslinked with 1% (v/v) formaldehyde for 10 min followed by quenching the crosslinking by the addition of 125 mM glycine. Genomic DNA extraction was performed using a MasterPure Yeast DNA Purification Kit (Epicenter). Starting with 2 µg, twofold serial dilution of the gDNA were prepared and spotted on a nylon membrane pre-equilibrate with PBS. After two washes with PBS the membrane was cross-linked in a UV-crosslinker (254 nm) at 120 J m⁻² for 10–15 s. After blocking (2% (w/v) BSA in PBS) the membrane was incubated with 2 µg/ml BG4 for 2 h at RT in agitation. Three washes with 0.1% (w/v) Tween in PBS were followed by 1 h incubation with 1:800 FLAG-Tag Antibody (Cell Signaling). Three washes with 0.1% (w/v) Tween/PBS were followed by 1 h incubation with 1:5000 Anti-HRP antibody (Santa Cruz). All antibodies were diluted in Blocking Buffer. The membrane was scanned by a ChemiDoc™ Gel Imaging System (BioRad)

**Gross chromosomal rearrangement assay**. The GCR assay was performed according to a published protocol[22]. Briefly, seven yeast cultures per GCR strain were grown at 30 °C for 48 h to saturation. $1 \times 10^{-7}$ cells diluted in water were plated on reference (YPD) or selective plates (drop-out medium lacking uracil and arginine (US Biologicals) supplemented with 1 g l⁻¹ 5-FOA and 60 mg l⁻¹ canavanine sulfate (FOA + Can)). After incubation for 4 days colony formation was counted. GCR clones are colonies that grew on selective plates. GCR rate was calculated using the FALCOR web server and MMS maximum likelihood method.

**γH2AX western blot**. Asynchronous cultures were grown to an OD (660 nm) of 0.6 and collected by centrifugation. Proteins were extracted by standard TCA purification and separated by SDS PAGE and transferred on a membrane. Western Blot analysis was performed with an antibody directed against γH2AX (Abcam) and Act1 (Santa Cruz Biotechnology). Proteins were detected using an enhanced chemiluminescence system (GE healthcare) and visualized with a Gel Doc XR + system (Bio-Rad). The pictures were quantified using ImageJ.

**Statistical analyses**. Significance was calculated based on one-sided Student's $t$-test. Asterisks' indicate statistical significance in comparison with wildtype cells: $^*p < 0.05$, $^{**}p < 0.01$, $^{***}p < 0.001$, $^{****}p < 0.0001$. Plotted results were based on the average of $N = 3$ biologically independent experiments.

**Reporting summary**. Further information on research design is available in the Nature Research Reporting Summary linked to this article.

## Data availability

ChIP-seq data have been deposited in the National Center for Biotechnology Information (NCBI) Sequencing Read Archive under the accession number GSE149502. Additionally, Supplementary Data 2 lists all peaks of the ChIP-seq analysis and Supplementary Data 3 lists all genes that were up- or down-regulated at the microarray analysis. All data is available from the authors upon reasonable request.

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

## Acknowledgements

We thank Wesley Browne (University of Groningen) for his help with circular dichroic measurements; Diana Spierings and Nancy Halsema for support with ChIP-seq analysis; Katharina Wanzek, Theresa Zacheja, and Eike Schwindt for providing strains; Hinke Kazemier and Michaela Limmer for experimental support; Markus Sauer for careful reading of the manuscript. Research in the Paeschke laboratory is funded by an ERC Stg Grant (638988-G4DSB) as well as the Deutsche Forschungsgemeinschaft (DFG, German Research Foundation) under Germany's Excellence Strategy—EXC2151–390873048. Open access funding provided by Projekt DEAL.

## Author contributions

Conceptualization, K.P.; Methodology, A.DM., S.G., E.F.S., S.A.J., M.H.; Data analysis, A.DM., S.G., S.A.J., M.C., M.H.; Writing—Original draft, S.G., A.DM, S.A.J., K.P.; Writing—Review and editing, A.DM., S.A.J., K.P.; Funding acquisition, K.P.; Resources, K.P.; Supervision, K.P.

## Competing interests

The authors declare no competing interests.
