## [Peer Review File · Nature Communications]

Reviewers' Comments:

Reviewer #1:

Remarks to the Author:

In this manuscript De Magis et al. describe, for the first time, a functional protein (Zuo1) that promotes G-quadruplex formation to preserve genomic stability in yeast. This is the first example that describes a functional and positive biological effect mediated by G-quadruplex DNA secondary structures, which is highly novel. Furthermore, the conclusions made by the authors are the result of a systematic and rigorous series of biological experiments. Hence, there is no doubt that this manuscript should be considered for publication in Nature Communication.

I have, however, a couple of concerns that I feel the authors should address prior publication:

1) From the way the authors interpret their results and also as mentioned a couple of times in the manuscript, it seems that Zuo1 is a G-quadruplex "inducer" and that the formation of these secondary structures at certain genomic loci stimulates NER activity. Although the authors have demonstrated that Zuo1 can bind to G4s biophysically, it would be good to show that this protein could actually stimulate/induce G4-formation in a non-folded sequence, to support this hypothesis, this can be done for example by CD. Also if Zuo1 is a G4-inducer, why there is no increase of G4-binding peaks in the Zuo1-overexpressed experiment? The authors should discuss this.

2) The second concern is still related to the first one. Authors can rescue yeast growth in Zuo1 deficient cells by addition of a G4-ligand such as PhenDC3, which confirms the function of the protein to promote G4-formation. However, they still speculate that Zuo1 binding to G4 is key to recruit Rad23 but with no evidence to support this directly. Is this really the case? Or is simply Zuo1 creating G4s that recruit the NER pathway on their own right? This seems to be more likely, otherwise how could PhenDC3 rescue yeast growth in cells that are still deficient of Zuo1? If the protein binding to G4 is key to recruit NER PhenDC3 alone could not rescue the phenotype. This aspect should be discussed more and caveated in the manuscript prior publication.

Reviewer #2:

Remarks to the Author:

This work describes a potentially interesting discovery linking the protein Zuo1 to G4 structures and nucleotide excision repair. By discovering that Zuo1 recognizes G4 structures and allows the recruitment of NER proteins, the authors suggest that this mechanism allows the NER mechanism to work. The authors' results are convincing and clearly demonstrate that Zuo1 recognizes G4 and recruits NER proteins. It remains however that the most interesting part of the authors' model, that is to say the repair of UV lesions mediated by the G4 structure formation recognized by Zuo1, remains a hypothesis that is not demonstrated directly in this paper. The absence of this demonstration makes the paper less attractive, lacking a bit of consistency for a journal like Nature Communications.

I recognize that it may be difficult to demonstrate this hypothesis, but it would be interesting to know what fraction of UV damage lesions are repaired by this mechanism in a cell? Could the authors demonstrate a delay in eliminating UV lesions in the absence of Zuo1 (using Dot Blot or even better XR-seq approach)?

Minor details

1- To find the protein that directly binds to Zuo1 on G4 in order to recruit NER factor would increase the impact of the work.

2- The experiments of ChIP and DOT blot indicate that the absence of Zuo1 leads to a decrease amount of the G4 motifs. The authors conclude that Zuo1 supports their formation but can they exclude the hypothesis that the presence of Zuo1 simply prevents their resolutions?

3- I would advice to the authors to combine figure 3-b and 3-d to avoid repetition of the wt and Zuo1 lanes in these two panels.

4- It would be important to include sgs1d and pif1-m2 yeast strains alone in experiments 3-c-d rather than only refering to published papers

Reviewer #1 (Remarks to the Author):

In this manuscript De Magis et al. describe, for the first time, a functional protein (Zuo1) that promotes G-quadruplex formation to preserve genomic stability in yeast. This is the first example that describes a functional and positive biological effect mediated by G-quadruplex DNA secondary structures, which is highly novel. Furthermore, the conclusions made by the authors are the result of a systematic and rigorous series of biological experiments. Hence, there is no doubt that this manuscript should be considered for publication in Nature Communication.

I have, however, a couple of concerns that I feel the authors should address prior publication:

We are grateful to the positive and very constructive comments of both reviewer and for considering our manuscript for publication in Nature Communication.

1) From the way the authors interpret their results and also as mentioned a couple of times in the manuscript, it seems that Zuo1 is a G-quadruplex "inducer" and that the formation of these secondary structures at certain genomic loci stimulates NER activity. Although the authors have demonstrated that Zuo1 can bind to G4s biophysically, it would be good to show that this protein could actually stimulate/induce G4-formation in a non-folded sequence, to support this hypothesis, this can be done for example by CD. Also if Zuo1 is a G4-inducer, why there is no increase of G4-binding peaks in the Zuo1-overexpressed experiment? The authors should discuss this.

We thank the reviewer for this valid point. As suggested by the referee we performed CD titration experiments, with the purified Zuo1 and the G4 from Chromosome IX and its mutated version (G4mut). These experiments were done in the presence of 100 mM K+, to support G4 formation. Our CD measurements revealed an increase up to 15 mdeg of the maximum peak at 264 nm at 5 and 10 μ M (Supp. Fig. S1f).

The second comment of the referee is correct, we also expected to observe after Zuo1 overexpression more G4s. Regardless, we did not see more G4 after overexpression of Zuo1-oex experiments. We explain these observations in such manner that Zuo1 under normal conditions binds to a specific subset of G4s that do not increase after Zuo1 overexpression. Meaning after Zuo1-overexpression not more G4 can fold because no additional free G4 targets of Zuo1 are available. Also, we would like to note that stabilization does not mean G4 induction. We believe that these data further underline the specificity of Zuo1 to bind specifically to folded G4 structures. We have discussed this in the paper.

2) The second concern is still related to the first one. Authors can rescue yeast growth in Zuo1 deficient cells by addition of a G4-ligand such as PhenDC3, which confirms the function of the protein to promote G4-formation. However, they still speculate that Zuo1 binding to G4 is key to recruit Rad23 but with no evidence to support this directly. Is this really the case? Or is simply Zuo1 creating G4s that recruit the NER pathway on their own right? This seems to be more likely, otherwise how could PhenDC3 rescue yeast growth in cells that are still deficient of Zuo1? If the protein binding to G4 is key to recruit NER PhenDC3 alone could not rescue the phenotype. This aspect should be discussed more and caveated in the manuscript prior publication.

We thank the reviewer for this very valid comment, and agree that "Zuo1 can create G4s that recruit the NER pathway on their own right" and that "Zuo1 binding to G4 is key to recruit Rad23" is an overstatement. We have, as suggested, modified the results and discussion accordingly.

Reviewer #2 (Remarks to the Author):

This work describes a potentially interesting discovery linking the protein Zuo1 to G4 structures and nucleotide excision repair. By discovering that Zuo1 recognizes G4 structures and allows the recruitment of NER proteins, the authors suggest that this mechanism allows the NER mechanism to work. The authors' results are convincing and clearly demonstrate that Zuo1 recognizes G4 and recruits NER proteins. It remains however that the most interesting part of the authors' model, that is to say the repair of UV lesions mediated by the G4 structure formation recognized by Zuo1, remains a hypothesis that is not demonstrated directly in this paper. The absence of this demonstration makes the paper less attractive, lacking a bit of consistency for a journal like Nature Communications.

I recognize that it may be difficult to demonstrate this hypothesis, but it would be interesting to know what fraction of UV damage lesions are repaired by this mechanism in a cell? Could the authors demonstrate a delay in eliminating UV lesions in the absence of Zuo1 (using Dot Blot or even better XR-seq approach)?

We are grateful to the reviewer for his/her constructive comments. The suggested Dotblot would be a perfect experiment to address his question. We ordered in January the required antibody, but due to the current Covid-19 situation (antibody is produced in China and distributed via US), we have not received the antibody, yet. In the meantime, we have thought about alternative experiments, that are under the current situation doable and will address the reviewers concern.

We have performed, and included in the paper, a western blot (WB) analysis using an antibody directed against γ H2AX (marker of double strand break). Thymine dimers and other photo adducts are recognized by stochastic order assembly of RPA, XPA and XPC-TFIIH and then process as double strand break. We are using the timing as well as the decay of γ H2AX as a read out of damage and repair. We treated wildtype and *zuo1* Δ cells with UV and measured the levels of γ H2AX over time (after 0, 1 and 4 hours after UV exposure). In the WB a delay in eliminating UV lesions was detectable in *zuo1* Δ cells (Supp. Fig. 5a-b). In detail, 4 hours after treatment wildtype cells show nearly no γ H2AX signal. We interpret this as 100% repair of the UV lesions. In the *zuo1* Δ cells only 39.5% of the UV lesions were repaired. These data indicate that there is a 60% fraction that is normally repaired by the mechanism involving Zuo1 after UV irradiation.

Minor details

1- To find the protein that directly binds to Zuo1 on G4 in order to recruit NER factor would increase the impact of the work.

We thank the reviewer for the comment, and agree that if we find the protein that directly binds to Zuo1 at G4 regions in order to recruit NER factor would increase the impact of the work. However, the direct interaction partner is interesting, but since the recruitment of the NER machinery is mainly done by G4 structures and only supported by Zuo1 (see referee comment 1) the identification of the interaction partner is interesting but beyond the scope of this publication. We added a to the discussion the relevance of G4 itself for NER protein recruitment.

Regardless, the most straight forward approach would be a co-IP followed by mass spectrometry. We performed a co-IP against Zuo1 but identified mainly ribosomal proteins. This we explain by published findings that Zuo1 is also described as a ribosome-associated factor, which leads to many ribosomal proteins in this approach (data not shown, because not relevant). A cellular fractionation, which would allow to distinguish between cytosolic or nuclear interaction partner is not really doable in yeast.

2- The experiments of ChIP and DOT blot indicate that the absence of Zuo1 leads to a decrease amount of the G4 motifs. The authors conclude that Zuo1 supports their formation but can they exclude the hypothesis that the presence of Zuo1 simply prevents their resolutions?

We thank the reviewer for the comment, and agree that there is the possibility that “the presence of Zuo1 simply prevents their resolutions”. However, in yeast there are two main helicases known that unwind G4: Sgs1 and Pif1. If Zuo1 prevent G4 unfolding, we expected that in the absence of Zuo1 these helicases should bind better to the G4 regions. However, we observed in Fig. 3e, that both Sgs1 and Pif1 binding to G4 targets is reduced in *zuo1Δ* strain. Indicating that if Zuo1 prevents G4 unfolding this is not mediated by Sgs1 nor Pif1. In addition, we included in the new version CD titration analysis (see comment referee 1) in which we revealed that Zuo1 supports G4 stabilization in the cells.

3- I would advice to the authors to combine figure 3-b and 3-d to avoid repetition of the wt and Zuo1 lanes in these two panels.

For easier reading we originally decided to repeat *zuo1Δ* strain in both figures. But as suggested we removed *zuo1Δ* lanes from fig 3d in order to avoid repetition, but we decided to keep as 2 different figures in order to have the straight forward flow of the text.

4- It would be important to include *sgs1d* and *pif1-m2* yeast strains alone in experiments 3-c-d rather than only referring to published papers

We agree that referring to published papers is not enough. For this reason, we had already included the full growth rates in Supp. Fig. 3. In the text at page 7 lines 5 and 6 is reported: “(Supplementary Fig. 3b, c shows the growth rates of single and double mutants)”

Reviewers' Comments:

Reviewer #1:

Remarks to the Author:

The authors have fully addressed my concerns in their revised version of the manuscript and I am now fully supportive of publication in Nature Communications

Reviewer #2:

Remarks to the Author:

Considering the actual pandemic situation and the difficulty to get access to biological material such as antibodies I acknowledge the efforts of the authors who have try to answer to my queries, I am not sure that gammaH2AX is the perfect marker to measure NER efficacy and the paper would have been much stronger with proper approaches but I guess that the experiments performed by the authors is probably a beginning of answer and I agree that the paper deserved publication in Nature Communications

Reviewer #1 (Remarks to the Author):

The authors have fully addressed my concerns in their revised version of the manuscript and I am now fully supportive of publication in Nature Communications

We thank the reviewer for this nice comment

Reviewer #2 (Remarks to the Author):

Considering the actual pandemic situation and the difficulty to get access to biological material such as antibodies I acknowledge the efforts of the authors who have try to answer to my queries, I am not sure that gammaH2AX is the perfect marker to measure NER efficacy and the paper would have been much stronger with proper approaches but I guess that the experiments performed by the authors is probably a beginning of answer and I agree that the paper deserved publication in Nature Communications

We would like to thank Reviewer #2 for his understanding and nice words.